# Ceramic Stress Sensor Based on Thick Film Piezo-Resistive Ink for Structural Applications

**DOI:** 10.3390/s24020599

**Published:** 2024-01-17

**Authors:** Gabriele Bertagnoli, Mohammad Abbasi Gavarti, Mario Ferrara

**Affiliations:** 1Department of Structural, Geotechnical and Building Engineering (DISEG), Politecnico di Torino, Corso Duca degli Abruzzi, 24, 10129 Turin, Italy; mario.ferrara@polito.it; 2Department of Mechanical Engineering, Politecnico di Milano, Via La Masa, 34, 20156 Milan, Italy; mohammad.abbasi@polimi.it

**Keywords:** stress sensor, ceramic, piezoresistive, thick film, concrete, masonry

## Abstract

This paper presents a ceramic stress sensor with the dimension of a coin, able to measure the compressive force (stress) applied to its two round faces. The sensor is designed and engineered to be embedded inside concrete or masonry structures, like bridges or buildings. It provides good accuracy, robustness, and simplicity of use at potentially low cost for large-scale applications in civil structures. Moreover, it can be calibrated temperature compensated, and it is inherently hermetic, ensuring the protection of sensitive elements from the external environment. It is, therefore, suitable for operating in harsh and dirty environments like civil constructions. The sensor directly measures the internal stress of the structure, exploiting the piezo resistivity of thick film ink based on ruthenium oxide. It is insensitive with respect to the stiffness of the embedding material and the variation of the surrounding material properties like concrete hardening, shrinkage, and creep as it decouples the two components of stress.

## 1. Introduction

All civil structures are subjected to aging and deterioration, and their monitoring and damage evaluation/identification have become of major importance [1]. Many structures are nowadays equipped with sensors for Structural Health Monitoring (SHM) to assess their structural integrity and/or performance or to control the external loads applied to them. Optical sensors and accelerometers are the most used devices for SHM [2]. However, it has been shown that stress sensors could improve the reliability of monitoring systems as well as the accuracy of damage identification [3]. Stress sensors should have a low cost to be widely spread within the structure. For this reason, Micro-Electro-Mechanical Systems (MEMES) technology is the most suitable. Nevertheless, existing MEMES stress sensors have very limited full-scale applications, and they are unable to separate the contributions of normal and shear stresses at the material-package interface.

Measuring stress within a solid body presents considerable difficulties. The measure is usually obtained indirectly by measuring strains on the outer surface of the elements of the structure or directly within the structure. The measurement of a strain is converted into the measurement of stress by knowing in advance the constitutive equation of the material being measured. This conversion is simple and reliable for linear elastic materials, whose mechanical properties are constant over time and uniform in space within the structure itself, for example, metals like steel and aluminum alloys.

Major difficulties occur when it is desired to measure stress within a structure in which the material characteristics are neither uniform in space nor constant over time and are generally not precisely known in advance, as is the case of all cementitious materials like concrete and mortars. 

An additional complication is provided by the viscoelastic nature of cementitious materials, which leads to non-constant deformations and stress states in time, even when constant loads are applied [4,5]. 

For this reason, the sensor presented in this paper directly measures the stresses within the structure and not its deformation. The sensor can work in both static and dynamic/cyclic applications. It was tested in the laboratory in quasi-static conditions (load applied with a maximum speed of 0.1 MPa/s). The sampling frequency used during all laboratory tests was 100 Hz, but sampling up to 200 Hz is possible.

The resolution of the device is 5 N (0.01 MPa), but its accuracy is about 50 N (0.1 MPa).

Concrete is generally considered a homogeneous material at the scale of centimeters, but it is a heterogeneous material at the scale of millimeters. It can be seen as a mix of natural stones (sand and gravel with a diameter of 2 to 20 mm), a cement paste made of cement water and finer aggregates (diameter < 1 mm), and air bubbles. Such heterogeneity generates spatial variability in the stress to be measured even under a perfectly uniformly distributed external load.

If the sensor is close to big aggregates (stones with 10 to 20 mm of diameter), it will measure a higher stress than the average one. If it is close to air bubbles and cement paste, it will measure stress lower than the average one applied. Such variability is estimated by the authors to reach 0.5 MPa; therefore, the accuracy of the sensor within concrete is not a function of the sensor itself but mostly depends on the embedding material heterogeneity. The authors consider 0.5 MPa to be a good estimation of the real operative accuracy.

The sensor has been tested in an oil bath from 0 to 25 MPa of pressure, showing linear output and perfect working conditions. Nevertheless, all tests within concrete specimens have been performed with a maximum pressure of 10 MPa. The reason is that many concrete structures (like residential buildings) are subjected to stresses between 0 and 10 MPa in serviceability conditions.

The sensor is also thermally compensated. The first intrinsic thermal compensation is obtained by means of Wheatstone bridges. The finer temperature calibration process is carried out during production by testing the response at three different temperatures.

## 2. State of the Art

Ceramic stress sensors have become more and more used in SHM in recent years. A review of their use and future perspectives of these sensors can be found in [6,7]. Gu et al. [8], Song et al. [9], Laskar et al. [10], and Kong et al. [11] developed an advanced multifunctional PZT-based sensor, smart aggregate (SA) to diagnose the structural health of RC structures. Liao et al. [12] used pre-embedded piezoceramic-based sensors to measure dynamic stress and perform the SHM of a concrete column.

Nevertheless, ceramic sensors are not widely spread and are used for concrete or masonry SHM. Therefore, the solutions most frequently adopted by practitioners for the investigation of concrete or masonry structures will be presented in this section:(a)flat jacks;(b)deformation meters;(c)concrete stress meters.

### 2.1. Flat Jacks

Flat Jacks are formed using a flattened shell, for example, two sheets of steel welded at the perimeter, containing a fluid (generally oil) of which the pressure is measured by a manometer. Flat jacks typically have a wide, thin shape and are commonly only used for single (not continuous) measurements. They are temporarily inserted into specific cavities formed in concrete or masonry walls or columns [13,14]. Once the investigation is concluded, they are generally extracted from the structure, and the cavities are filled with mortar or similar materials to restore the structural integrity.

The stress acting on the structure in the direction orthogonal to the faces of the jack is obtained by recording the value of the pressure of the fluid by a set of hydraulic connections and reading devices.

This technical solution has several drawbacks:it requires demolition interventions to form the housings for the jacks within the structure;the dimensions of the steel sheets, of several tens of square centimeters, maybe too invasive for an element with small dimensions and may form weak points endangering the structural safety;this is a solution typically used in the presence of a human operator who executes the installation, takes the measurements, removes the instruments, and checks the restoration of the structural damage introduced;it is not a set of tools typically suitable for being permanently connected to a structure under investigation since the pressure of the fluid would also have to be kept constant over time at an extremely low tolerance, and the system of hydraulic connections is often incompatible with permanent applications.

### 2.2. Deformation Meters

Deformation meters are tools suitable for measuring variations in the distance between two points forming the measurement base. They may be mechanical, electromechanical, electrical, inductive and/or magnetic, optical, or laser devices.

The measurement base is generally positioned on the outer faces of the structure to be monitored and may be of a length ranging from a few millimeters to tens of centimeters.

Depending on the technology, the tools measure the number of rotations of a gear system, variations in electrical resistance in a magnetic field, wavelength, or the number of waves of a light source, or in the electrical capacitance of a capacitor.

All the measurements are subsequently traced back to the relative displacement between the two points forming the measurement base. The average deformation on the measurement base can thus be obtained by dividing the relative displacement by the initial length of the measurement base. The stress can subsequently be derived from the deformation if the constitutive equation of the material is known.

Technical solutions of this type have two major drawbacks:the mechanical features of the cementitious mixes, such as the constitutive equation, are not constant either in space or over time and are not generally precisely known in advance;cementitious mixes are viscoelastic; in other words, the deformations vary considerably over time (even by amounts greater than 200%), even in the presence of constant stresses, and the viscoelastic equation governing this variation is not generally precisely known in advance.

Some applications of strain meters within concrete may be found in the research done by Riley et al. [15] and Nield et al. [16].

A specific detection system for monitoring strains within concrete is described in U.S. Pat. No. 3286513 A [17]. It is a concrete probe intended to be incorporated into a concrete element for simultaneously measuring six deformations at a desired point within the concrete casting.

The elements responsible for measuring the deformations are arranged in a tetrahedral shape, and the extensometers used are glued to said tetrahedral support structure. The six detected deformations are related to six different local contributions.

The extensometers mounted on the tetrahedral structure measure the deformations of the agglomerate and thus provide data affected by the viscosity of the material. They do not provide accurate information about the stress state in the agglomerate.

### 2.3. Concrete Stress-Meters

Another type of device is concrete stress meters: an example of these products is given by [18]. These tools can be inserted inside the concrete casting. They are made of a porous cup-shaped shell within which a portion of agglomerate equivalent to that of the structure under investigation is held. The stresses of the enclosed agglomerate are read by a load cell housed at the base of said cup.

This solution has the following drawbacks:This is a bulky investigation system (the stress meter has a cylindrical structure of a length of approximately 600 mm and a diameter of approximately 76 mm) and thus requires sufficiently large structures to be able to house it, making it only applicable to castings having large dimensions, in which introducing the device does not lead to significant interference in the structural behavior.The stress meter has to be suitably connected within the cementitious structure under investigation during the casting steps and has to be positioned immediately before casting and filled with the same material forming the casting immediately before being installed, thus interfering with the other construction site operations.Given the dimensions of the container, suitable for inserting inert constituents of the agglomerate into it, the measurement obtained is not point information but rather an average over a length of 600 mm. The device, thus cannot be used in the case of sensitive stress gradients that affect the dimensions of the device.

## 3. Sensor Geometry and Working Principle

The properties of piezoelasticity have already been used to develop force sensors to be used for concrete elements, as shown in [19].

In this paper is presented a multi-axial stress sensor based on thick film piezoresistive ink [20] (see Figure 1). It can measure both the strains orthogonal to the round faces of the coin (called out-of-plane strains) and the radial strains parallel to the round faces of the coin (called in-plane strains). The stress orthogonal to the round faces of the device is then derived.

The sensor is insensitive with respect to the variation of the surrounding material properties because it can decouple the two components of strain. It can, therefore, overcome the issues of unknown modulus of elasticity and viscosity of the embedding material described in the previous paragraphs.

The sensor working principle is presented in this paper. A comparison, for different confinement conditions, between deformations calculated through closed-form formulations and through numerical simulations with the f.e.m. model is also reported. 

The proposed analytical solutions explore a wide range of ratios between applied axial pressures (orthogonal to the round faces of the coin) that should be directly proportional to the applied load to be measured and radial confinement stress that depends on boundary conditions and embedding material properties.

If a constant axial load is applied to the concrete element where the sensor is placed, the axial pressure on the sensor does not change, but the strains in concrete can significantly vary (2 to 3 times) because of creep. Creep and shrinkage, like other variations of material properties in time (aging, damage, cracking, etc.), can change the confinement (radial) action, but the axial stress remains proportional to the applied load because of equilibrium equations.

Results of laboratory tests on this sensor, directly loaded or embedded in concrete elements, can be found in [21,22]. Applications of this sensor to masonry structures can be found in [23].

The exact dimensions of the sensor are related to the intended application. 

Concrete is a non-uniform material containing particles (such as gravel, sand, voids, etc.) of a non-negligible size that may give rise to local stress variations. Therefore, the diameter should have a dimension big enough to average the irregularities but small enough not to create a dangerous discontinuity within the structure. 

The diameter can therefore vary between 1.5 and 4 cm. The total thickness of the sensor should be as slim as possible to limit the discontinuity created by the sensor itself within the stress field in the structure. In the proposed application, it is approximately one-tenth of the diameter of the bases (see Figure 1).

The body of the sensor is made of three layers of Aluminum Oxide Al_2_O_3_. The two external plates are thicker than the internal ones to provide mechanical protection to the sensing devices. The central layer is glued to the external ones by means of two layers of glass frit bonding. The geometry of the sensor is shown in Figure 2.

The top surface of the middle layer contains the piezoresistive gauges that are connected to form two Wheatstone bridges: one bridge senses in-plane strain, and the other one senses both in-plane and out-of-plane strain [24,25]. The design of the sensor and the position of the sensing elements were optimized by means of finite element analyses.

The Wheatstone bridges are realized on the upper face of the intermediate ceramic layer Figure 2c and embedded inside the top glass frit Figure 2d. The bridges are called Planar (*PL*) bridges and three-dimensional (3*D*) bridges, and are shown in Figure 3. Moreover, each sensor is made of the main resistor and secondary resistor, which represent the calibration and compensation resistor.

Each bridge is realized by four main resistors and two pairs of calibration resistors. In this paper, reference will be made to a theoretical bridge in which only four theoretical resistors (*R*_1_, *R*_2_, *R*_3_, *R*_4_) are considered; each of these four resistors contains the contribution of both measuring resistors and calibration resistors. The presence of resistors PT8, PT9, PT8A, PT9A, R6, R7, R6A, and R7A, shown in Figure 3, is therefore neglected in this work.

### 3.1. Wheatstone Bridge Working Principle

Given a Wheatstone bridge shown in Figure 4, the relation between the measured output $V_{out,0}$ and the input signal $V_{in}$ is given by the following equation:(1)$$\frac{V_{out,0}}{V_{in}}=\frac{R_{1}R_{3}-R_{2}R_{4}}{\left(R_{1}+R_{2}\right)\left(R_{3}+R_{4}\right)}$$

After deformation, the output variation between the undeformed state and the deformed state is given by the following relation:(2)$$\frac{{\Delta V}_{out}}{V_{in}}=\frac{V_{out,def}-V_{out,0}}{V_{in}}=\frac{\left(R_{1}+\Delta R_{1}\right)\left(R_{3}+\Delta R_{3}\right)-R_{2}R_{4}}{\left(R_{1}+\Delta R_{1}+R_{2}\right)\left(R_{3}+\Delta R_{3}+R_{4}\right)}-\frac{R_{1}R_{3}-R_{2}R_{4}}{\left(R_{1}+R_{2}\right)\left(R_{3}+R_{4}\right)}$$
where the term $\frac{R_{1}R_{3}-R_{2}R_{4}}{\left(R_{1}+R_{2}\right)\left(R_{3}+R_{4}\right)}$ does not depend on the deformation but only on the bridge electric resistors.

### 3.2. Characteristics of the Wheatstone Bridges Embedded in the Ceramic Sensor

The generic scheme of the Wheatstone bridge illustrated in Figure 4 has been particularized for both planar and 3*D* bridges, as shown in Figure 5, where:

Planar bridge:$R_{1}=R_{PL}$ in Figure 5a corresponds to R1A shown in Figure 3.$R_{3}=R_{PL}$ in Figure 5a corresponds to R3A shown in Figure 3.$R_{2}=R_{zero1}$ in Figure 5a corresponds to R2A shown in Figure 3.$R_{4}=R_{zero1}$ in Figure 5a corresponds to R4A shown in Figure 3.

3*D* bridge:$R_{1}=R_{3D}$ in Figure 5b corresponds to R1−1, R1−2, R1−3, R1−4 assembled in series parallel as shown in Figure 3.$R_{3}=R_{3D}$ in Figure 5b corresponds to R3−1, R3−2, R3−3, R3−4 assembled in series parallel as shown in Figure 3.$R_{2}=R_{zero2}$ in Figure 5b corresponds to R2 shown in Figure 3.$R_{4}=R_{zero2}$ in Figure 5b corresponds to R4 shown in Figure 3.

Theoretical resistors $R_{1},\;R_{2},\;R_{3},\;R_{4}$ in both bridges have values ranging between 14 and 16 kΩ. However, after completing the sensor production, it is not possible to measure their values. For this reason, to simplify the following equations, it has been assumed that:(3)$$R_{1}=R_{2}=R_{3}=R_{4}$$

This means that if resistor values are not equal, the initial ΔV value will be different from 0 and will coincide with the resistor offset. Therefore, if $R_{1}=R_{3}$:(4)$${\Delta R}_{1}=\Delta R_{3}=\Delta R$$

The variation of output potential can be found by substituting Equations (3) and (4) in Equation (2):(5)$$\frac{{\Delta V}_{out}}{V_{in}}=\frac{R^{2}+2R\Delta R+\Delta R^{2}-R^{2}}{4R^{2}+\Delta R^{2}+4R\Delta R}$$

The term ΔR is small if compared to R as will be seen in the next paragraph. Therefore $\Delta R^{2}$ is a very small value compared with the other elements, so it can be neglected. The resulting equation is:(6)$$\frac{{\Delta V}_{out}}{V_{in}}=\frac{\Delta R}{2R+2\Delta R}$$

Starting from this equation, it is possible to explicit the relation between ΔR and R for both planar and 3*D* bridges:(7)$$\frac{\Delta R_{PL,3D}}{R_{PL,3D}}=\frac{2\frac{\Delta V_{outPL,3D}}{V_{in}}}{1-2\frac{\Delta V_{outPL,3D}}{V_{in}}}$$

### 3.3. Resistors Piezo-Elastic Behavior

With reference to a simple resistor, it is possible to identify a local reference system in which the *x*-axis corresponds to the electric current flow, the *y*-axis is orthogonal to the *x*-axis and belongs to the plane on which the resistor is inked, and the *z*-axis is located outside of this plane, as shown in Figure 6.

The dimensions of the resistors used in the sensor are:(8)$$l_{x}=600\;\mu m,\;l_{y}=500\;\mu m,\;l_{z}=10\;\mu m$$

The variation ΔR of the resistance R due to the deformation of the resistor can be calculated according to the second Ohm law as follows:(9)$$\Delta R=\Delta \rho \frac{l_{x}}{S}+\Delta l_{x}\frac{\rho }{S}-\frac{\rho l_{x}\Delta S}{S^{2}}$$
where ρ is the resistivity of the material, $l_{x}$ is the length, and S is the cross-section of the resistor.

Therefore, the relative variation of the resistance is obtained by dividing Equation (9) by the original resistance as follows:(10)$$\frac{\Delta R}{R}=\frac{\Delta \rho \frac{l_{x}}{S}+\Delta l_{x}\frac{\rho }{S}-\frac{\rho l_{x}\Delta S}{S^{2}}}{\rho \frac{l_{x}}{S}}=\frac{\Delta \rho }{\rho }+\frac{\Delta l_{x}}{l_{x}}-\frac{\Delta S}{S}$$

The term $\frac{\Delta l_{x}}{l_{x}}$ is the strain $\varepsilon_{x}$.

The term $\frac{\Delta S}{S}$ can be calculated as follows in the function of the strains $\varepsilon_{y}$ and $\varepsilon_{z}$.
(11)$$\frac{\Delta S}{S}=\frac{S_{def}-S}{S}=\frac{\left(l_{y}+\Delta l_{y}\right)\left(l_{z}+\Delta l_{z}\right)-l_{y}l_{z}}{l_{y}l_{z}}=\varepsilon_{y}+\varepsilon_{z}+\varepsilon_{y}\varepsilon_{z}$$

Being $\varepsilon_{y}\varepsilon_{z}$ much smaller than the other terms it can be neglected obtaining:(12)$$\frac{\Delta S}{S}=\varepsilon_{y}+\varepsilon_{z}$$

The term $\frac{\Delta \rho }{\rho }$ is close to zero for common metals (like copper) but becomes a function of the deformation strains or stresses in piezoresistive materials (like silicon and germanium). It can be written as a function of a general state of strain following the next steps.

The first Ohm law can be written in vector form as follows:(13)$$\Delta V={\left\{E\right\}}^{T}\cdot \left\{\Delta l\right\}\;\to \;\left\{\begin{array}{ccc} E_{x} & E_{y} & E_{z} \end{array}\right\}\times \left\{\begin{array}{c} l_{x}\\ l_{y}\\ l_{z} \end{array}\right\}$$
(14)$$\left\{E\right\}=\left[\rho \right]\cdot \left\{j\right\}\to \left\{\begin{array}{c} E_{x}\\ E_{y}\\ E_{z} \end{array}\right\}=\left[\begin{array}{ccc} \rho_{xx} & \rho_{xy} & \rho_{xz}\\ & \rho_{yy} & \rho_{yz}\\ symm & & \rho_{zz} \end{array}\right]\times \left\{\begin{array}{c} j_{x}\\ j_{y}\\ j_{z} \end{array}\right\}$$
where $\left[\rho \right]$ is the resistivity tensor with a maximum of six different terms, $\left\{j\right\}$ is the density current vector and $\left\{E\right\}$ is the electrical field. If the material is homogeneous from the electrical point of view, $\left\{E\right\}$ and $\left\{j\right\}$ are parallel, and the resistivity is a scalar $\left[\rho_{ij}\right]\equiv \rho$.

The variation of each of the six $\rho_{ij}$ in function of the strain state reads [26]:(15)$$\left\{\frac{\Delta \rho_{ij}}{\rho }\right\}={\left[\Pi \right]}_{\varepsilon }\cdot \left\{\varepsilon_{ij}\right\}\to \frac{1}{\rho }\left\{\begin{array}{c} \Delta \rho_{xx}\\ \Delta \rho_{yy}\\ \Delta \rho_{zz}\\ \Delta \rho_{xy}\\ \Delta \rho_{xz}\\ \Delta \rho_{yz} \end{array}\right\}=\left[\begin{array}{cccccc} p_{11} & p_{12} & p_{12} & 0 & 0 & 0\\ & p_{11} & p_{12} & 0 & 0 & 0\\ & & p_{11} & 0 & 0 & 0\\ & & & p_{44} & 0 & 0\\ & & & & p_{44} & 0\\ symm. & & & & & p_{44} \end{array}\right]\times \left\{\begin{array}{c} \varepsilon_{x}\\ \varepsilon_{y}\\ \varepsilon_{z}\\ \gamma_{xy}\\ \gamma_{xz}\\ \gamma_{xz} \end{array}\right\}$$

In the case presented in this paper, only differential of potential along x direction and only current in x-direction are present, therefore:(16)$$\left\{E\right\}=\left\{\begin{array}{c} E_{x}=\Delta V/l_{x}\\ E_{y}=0\\ E_{z}=0 \end{array}\right\}\left\{j\right\}=\left\{\begin{array}{c} j_{x}\\ j_{y}=0\\ j_{z}=0 \end{array}\right\}$$
(17)$$E_{x}=\rho_{xx}j_{x}$$

And therefore $\frac{\Delta \rho }{\rho }$ becomes:(18)$$\frac{\Delta \rho_{xx}}{\rho }=p_{11}\varepsilon_{x}+p_{12}\varepsilon_{y}+p_{12}\varepsilon_{z}$$

Substituting Equations (18) and (12) into Equation (10) we get:(19)$$\begin{array}{c} \frac{\Delta R}{R}=\frac{\Delta \rho }{\rho }+\frac{\Delta l_{x}}{l_{x}}-\frac{\Delta S}{S}=p_{11}\varepsilon_{x}+p_{12}\varepsilon_{y}+p_{12}\varepsilon_{z}+\varepsilon_{x}-\varepsilon_{y}-\varepsilon_{z}\\ \frac{\Delta R}{R}=\left(p_{11}+1\right)\varepsilon_{x}+\left(p_{12}-1\right)\varepsilon_{y}+\left(p_{12}-1\right)\varepsilon_{z} \end{array}$$

That can be written in a compact form as:(20)$$\;\frac{\Delta R}{R}=G_{x}\varepsilon_{x}+G_{y}\varepsilon_{y}+G_{z}\varepsilon_{z}$$
where each $G_{i}$ coefficient is a function of a piezo-elastic term ($p_{11}$ and $p_{12}$) and a geometric term ($G_{iG}$). The geometric terms ($G_{xG}$, $G_{yG}$, $G_{zG}$) are equal to ±1 and express the relationship between the variation of electrical resistance and the strains along x, y, z for a non-piezo material (i.e., common copper).

For ruthenium oxide ink:(21)$$p_{11}=p_{12}=16.4$$
therefore Equation (19) becomes:(22)$$\frac{\Delta R}{R}=17.4\varepsilon_{x}+15.4\varepsilon_{y}+15.4\varepsilon_{z}$$

## 4. Simulation of Deformations Imposed to the Sensor

The sensor feels the loads applied to concrete or masonry structures as imposed deformations of the surrounding environment, not being able to change the overall force distributions in the structure because of its small dimensions. 

Some closed-form solutions for limit and typical cases of imposed deformations are derived in this paragraph. 

The sensor is approximated to be axial-symmetric and homogeneous as if it were only made of ceramic, neglecting the presence of the glass layer and the cavities. The corresponding global reference system in axial symmetry is illustrated in Figure 7.

Imposed deformations will be applied to the sensor, and the corresponding following parameters will be calculated:Average stresses in the sensor $\sigma_{v}$, $\sigma_{r}$, $\sigma_{\theta }$.Strains to which are subjected the resistors of *PL* and 3*D* bridge.Variation of resistances $\frac{\Delta R_{3D}}{R_{3D}}$ and $\frac{\Delta R_{PL}}{R_{PL}}$.Variation of output potential $\frac{\Delta V_{out}}{V_{in}}=\frac{\Delta R}{2R+2\Delta R}$ for both 3*D* and *PL* bridges.

Five different scenarios are analyzed: they are characterized by the same axial imposed strain $\varepsilon_{v}<0,$ and by different radial strains corresponding to different levels of lateral confinement or stretching as follows:(1).Nil radial expansion (∞ lateral confinement) corresponding to an equivalent Poisson ratio $\upsilon_{EQ}=0.00$. This scenario is a theoretical limit case that is almost impossible to achieve in a laboratory test case. It is, therefore, examined as a limit condition.(2).Confined radial expansion corresponds to an equivalent Poisson ratio smaller than the sensor’s one $\upsilon_{EQ}=0.10,$ meaning the sensor is confined in a stiffer material.(3).Free radial expansion (0 lateral confinement) corresponding to an equivalent Poisson ratio equal to the sensor’s one $\upsilon_{EQ}=\upsilon_{sensor}=0.20$.(4).Increased radial expansion $\upsilon_{EQ}=0.30$, corresponding to the sensor being encased in a material more deformable than ceramic.(5).Increased lateral expansion $\upsilon_{EQ}=0.56$, to match young concrete deformability used in f.e.m. simulations.

### 4.1. General Case with Imposed Lateral Expansion $\upsilon_{EQ}$

In this case, a constant vertical negative deformation is applied, and a given lateral expansion of the sensor is allowed.
(23)$$\left\{\begin{array}{c} \varepsilon_{v}<\;0\\ \varepsilon_{r}=\;-\upsilon_{EQ}\varepsilon_{v}\\ \varepsilon_{\theta }=\;-\upsilon_{EQ}\varepsilon_{v} \end{array}\right.$$

The average stresses corresponding to this imposed deformation are given by the axial-symmetric constitutive law, where the shear deformation is neglected since deformations imposed by Equation (23) are constant in the sensor:(24)$$\left\{\begin{array}{c} \sigma_{v}\\ \sigma_{r}\\ \sigma_{\theta } \end{array}\right\}=\frac{E}{\left(1+\nu \right)\left(1-2\nu \right)}\left[\begin{array}{ccc} 1-\nu & \nu & \nu \\ \nu & 1-\nu & \nu \\ \nu & \nu & 1-\nu \end{array}\right]\times \left\{\begin{array}{c} \varepsilon_{v}\\ -\upsilon_{EQ}\cdot \varepsilon_{v}\\ -\upsilon_{EQ}\cdot \varepsilon_{v} \end{array}\right\}$$
where E and ν can be considered for the whole sensor equal to the ones of ceramic accepting a small approximation ($E_{sensor}=250\;GPa$, $\nu_{sensor}=0.2$). Therefore:(25)$$\begin{array}{c} \sigma_{v}=\frac{E_{sensor}}{\left(1+\nu \right)\left(1-2\nu \right)}\left(1-\nu -2\nu \cdot \nu_{EQ}\right)\varepsilon_{v}\\ \sigma_{r}=\sigma_{\theta }=\frac{E_{sensor}}{\left(1+\nu \right)\left(1-2\nu \right)}\left[\nu -\nu_{EQ}\right]\varepsilon_{v} \end{array}$$

The relation between stress and strains in the glass layer containing the 3*D* bridge resistor is given by.
(26)$$\left\{\begin{array}{c} \sigma_{g,v}\\ \sigma_{g,r}\\ \sigma_{g,\theta } \end{array}\right\}=\frac{E_{g}}{\left(1+\nu_{g}\right)\left(1-2\nu_{g}\right)}\left[\begin{array}{ccc} 1-\nu_{g} & \nu_{g} & \nu_{g}\\ \nu & 1-\nu_{g} & \nu_{g}\\ \nu_{g} & \nu_{g} & 1-\nu_{g} \end{array}\right]\times \left\{\begin{array}{c} \varepsilon_{g,v}\\ \varepsilon_{g,r}\\ \varepsilon_{g,\theta } \end{array}\right\}$$
where:$E_{g}=75\;GPa$ and $\nu_{g}=0.18$ are the modulus of elasticity and the Poisson coefficient of glass frit.$\varepsilon_{g,r}=\varepsilon_{r}$ and $\varepsilon_{g,\theta }=\varepsilon_{\theta }$ are the radial and tangential strains in the glass frit that are equal to the ones of the sensor.$\sigma_{g,v}=\sigma_{v}$ is the vertical stress in the glass frit, equal to the one of the sensor.$\varepsilon_{g,v}$, $\sigma_{g,r}$, $\sigma_{g,\theta }$ are unknowns to be determined by solving the system written in Equation (26) as follows.
(27)$$\varepsilon_{g,v}=\frac{\frac{\sigma_{g,v}}{E_{g}}\left(1+\upsilon_{g}\right)\left(1-2\upsilon_{g}\right)-\upsilon_{g}\left(\varepsilon_{g,r}+\varepsilon_{g,\theta }\right)}{\left(1-\upsilon_{g}\right)}$$
(28)$$\sigma_{g,r}=\frac{E_{g}}{\left(1+\nu_{g}\right)\left(1-2\nu_{g}\right)}\left[\nu_{g}\varepsilon_{g,v}+\left(1-\upsilon_{g}\right)\varepsilon_{g,r}+\upsilon_{g}\varepsilon_{g,\theta }\right]$$
(29)$$\sigma_{g,\theta }=\frac{E_{g}}{\left(1+\nu_{g}\right)\left(1-2\nu_{g}\right)}\left[\nu_{g}\varepsilon_{g,v}+\upsilon_{g}\varepsilon_{g,r}+\left(1-\upsilon_{g}\right)\varepsilon_{g,\theta }\right]$$

By substituting in Equation (27): the value of $\sigma_{g,v}=\sigma_{v}$ given in Equation (25); $\varepsilon_{g,r}=\varepsilon_{r}=-\nu_{EQ}\varepsilon_{v}$ and $\varepsilon_{g,\theta }=\varepsilon_{\theta }=-\nu_{EQ}\varepsilon_{v}$ can be obtained.
(30)$$\varepsilon_{g,v}=\frac{\frac{\frac{E_{sensor}}{\left(1+\nu \right)\left(1-2\nu \right)}\left(1-\nu -2\nu \nu_{EQ}\right)\varepsilon_{v}}{E_{g}}\left(1+\upsilon_{g}\right)\left(1-2\upsilon_{g}\right)+2\upsilon_{g}\upsilon_{EQ}\varepsilon_{v}}{\left(1-\upsilon_{g}\right)}\;=\left[\frac{E_{sensor}}{E_{g}}\cdot \frac{\left(1-\nu -2\nu \nu_{EQ}\right)\left(1+\upsilon_{g}\right)\left(1-2\upsilon_{g}\right)}{\left(1+\nu \right)\left(1-2\nu \right)\left(1-\upsilon_{g}\right)}+\frac{2\upsilon_{g}\upsilon_{EQ}}{\left(1-\upsilon_{g}\right)}\right]\varepsilon_{v}\;=\left[\frac{250}{75}\cdot \frac{\left(1-0.2-0.4\nu_{EQ}\right)\left(1+0.18\right)\left(1-0.36\right)}{\left(1+0.2\right)\left(1-0.4\right)\left(1-0.18\right)}+\frac{0.36\upsilon_{EQ}}{\left(1-0.18\right)}\right]\varepsilon_{v}\;=\left(3.42-1.27\nu_{EQ}\right)\varepsilon_{v}$$

The resistors of the 3*D* and *PL* bridges undergo respectively the following deformations in their local referring systems:(31)$$3D\left\{\begin{array}{c} \varepsilon_{x}=\varepsilon_{y}=\varepsilon_{r}=\varepsilon_{\theta }=-\nu_{EQ}\varepsilon_{v}\\ \varepsilon_{z}=\varepsilon_{g,v} \end{array}\right.\;\;\;\;\;PL\left\{\begin{array}{c} \varepsilon_{x}=\varepsilon_{y}=\varepsilon_{r}=\varepsilon_{\theta }=-\nu_{EQ}\varepsilon_{v}\\ \varepsilon_{z}=-\upsilon_{PL}\varepsilon_{x}-\upsilon_{PL}\varepsilon_{y} \end{array}\right.$$
where $\upsilon_{PL}=\upsilon_{RuO}=0.28$ is the Poisson ratio of the ruthenium oxide.

The variation of the resistance of the two bridges can, therefore, be calculated using Equation (22) and the strains given by Equation (31).
(32)$$\begin{array}{c} \frac{\Delta R_{3D}}{R_{3D}}=-\left(G_{1}+G_{2}\right)\nu_{EQ}\varepsilon_{v}+G_{2}\varepsilon_{g,v}\\ \frac{\Delta R_{PL}}{R_{PL}}=-\left(G_{1}+G_{2}\right)\nu_{EQ}\varepsilon_{v}+G_{2}2\nu_{PL}\nu_{EQ}\varepsilon_{v} \end{array}$$

Introducing Equation (30) within the first term of Equation (32), we express the variation of the resistances as a pure function of the applied deformation expressed in terms of $\varepsilon_{v}$ and $\upsilon_{g}$ obtaining:(33)$$\begin{array}{c} \frac{\Delta R_{3D}}{R_{3D}}=\left\{-\left(G_{1}+G_{2}\right)\nu_{EQ}+G_{2}\left[\frac{E_{sensor}}{E_{g}}\times \frac{\left(1-\nu -2\nu \cdot \nu_{EQ}\right)\left(1+\upsilon_{g}\right)\left(1-2\upsilon_{g}\right)}{\left(1+\nu \right)\left(1-2\nu \right)\left(1-\upsilon_{g}\right)}+\frac{2\upsilon_{g}\upsilon_{EQ}}{\left(1-\upsilon_{g}\right)}\right]\right\}\varepsilon_{v}\\ \frac{\Delta R_{PL}}{R_{PL}}=\left[-\left(G_{1}+G_{2}\right)\nu_{EQ}+G_{2}2\nu_{PL}\nu_{EQ}\right]\varepsilon_{v} \end{array}$$

That can be written in compact form as:(34)$$\begin{array}{c} \Delta R_{3D}=A_{3D}R_{3D}\varepsilon_{v}\\ \Delta R_{PL}=A_{PL}R_{PL}\varepsilon_{v} \end{array}$$

And therefore, the variation of output potential is:(35)$$\begin{array}{c} 3D)\frac{\Delta V_{out}}{V_{in}}=\frac{\Delta R}{2R+2\Delta R}=\;\frac{A_{3D}R\varepsilon_{v}}{2R+2A_{3D}R\varepsilon_{v}}\\ PL)\frac{\Delta V_{out}}{V_{in}}=\frac{\Delta R}{2R+2\Delta R}=\;\frac{A_{PL}R\varepsilon_{v}}{2R+2A_{PL}R\varepsilon_{v}} \end{array}$$

The terms $2A_{3D,PL}R\varepsilon_{v}$ in the denominator of both Equation (35) can be neglected as it is much smaller than 2R, being $-1E-3<\varepsilon_{v}<\;0$ in the field of application presented herein; therefore Equation (35) becomes:(36)$$\begin{array}{c} 3D)\frac{\Delta V_{out}}{V_{in}}=\frac{A_{3D}\varepsilon_{v}}{2}\\ PL)\frac{\Delta V_{out}}{V_{in}}=\;\frac{A_{PL}\varepsilon_{v}}{2} \end{array}$$
and expressed in Equation (36) $\varepsilon_{v}$ in function of $\sigma_{v}$ as expressed in Equation (25), the variation of output potential can be calculated in function of the applied pressure $\sigma_{v}$.
(37)$$\begin{array}{c} 3D)\frac{\Delta V_{out}}{V_{in}}=\frac{A_{3D}}{2E_{sensor}}\frac{\left(1+\nu \right)\left(1-2\nu \right)}{1-\nu -2\nu \nu_{EQ}}\sigma_{v}\\ PL)\frac{\Delta V_{out}}{V_{in}}=\;\frac{A_{PL}}{2E_{sensor}}\frac{\left(1+\nu \right)\left(1-2\nu \right)}{1-\nu -2\nu \nu_{EQ}}\sigma_{v} \end{array}$$
where:(38)$$\begin{array}{c} A_{3D}=-\left(G_{1}+G_{2}\right)\nu_{EQ}+G_{2}\left[\frac{E_{sensor}}{E_{g}}\cdot \frac{\left(1-\nu -2\nu \nu_{EQ}\right)\left(1+\upsilon_{g}\right)\left(1-2\upsilon_{g}\right)}{\left(1+\nu \right)\left(1-2\nu \right)\left(1-\upsilon_{g}\right)}+\frac{2\upsilon_{g}\upsilon_{EQ}}{\left(1-\upsilon_{g}\right)}\right]\\ A_{PL}=-\left(G_{1}+G_{2}\right)\nu_{EQ}+G_{2}2\nu_{PL}\nu_{EQ} \end{array}$$
substituting in Equation (38) the numerical values of the physical entities we get:(39)$$\begin{array}{c} A_{3D}=52.67-52.35\nu_{EQ}\\ A_{PL}=-24.18\nu_{EQ} \end{array}$$

And therefore, Equation (37) becomes:(40)$$\begin{array}{c} 3D)\frac{\Delta V_{out}}{V_{in}}=\frac{52.67\;-\;52.35\nu_{EQ}}{2\;\times \;250}\frac{0.72}{0.8\;-\;0.4\nu_{EQ}}\sigma_{v}\left[\frac{mV}{V}\right]\\ PL)\frac{\Delta V_{out}}{V_{in}}=\;-\frac{24.18\nu_{EQ}}{2\;\times \;250}\frac{0.72}{0.8\;-\;0.4\nu_{EQ}}\sigma_{v}\left[\frac{mV}{V}\right] \end{array}$$

### 4.2. Nil Lateral Expansion $\upsilon_{EQ}=0.00$

In this case, a constant vertical negative deformation is applied, and nil lateral expansion of the sensor is allowed $\upsilon_{EQ}=0.00$, Equation (40) becomes:(41)$$\begin{array}{c} 3D)\frac{\Delta V_{out}}{V_{in}\cdot \sigma_{v}}=9.48\times 10^{-2}\left[\frac{mV}{V\cdot MPa}\right]\\ PL)\frac{\Delta V_{out}}{V_{in}\cdot \sigma_{v}}=\;0\left[\frac{mV}{V\cdot MPa}\right] \end{array}$$

### 4.3. Confined Lateral Expansion $\upsilon_{EQ}=0.10$

In this case, constant confinement has been imposed to allow a lateral expansion equal to 10% of vertical one ($\upsilon_{EQ}=0.10$); Equation (40) becomes:(42)$$\begin{array}{c} 3D)\frac{\Delta V_{out}}{V_{in}\cdot \sigma_{v}}=8.99\times 10^{-2}\left[\frac{mV}{V\cdot MPa}\right]\\ PL)\frac{\Delta V_{out}}{V_{in}\cdot \sigma_{v}}=\;-4.58\times 10^{-3}\left[\frac{mV}{V\cdot MPa}\right] \end{array}$$

### 4.4. Free Lateral Expansion $\upsilon_{EQ}=0.20$

In this case, the lateral expansion of the sensor will be allowed, and a compressive axial deformation will be imposed; Equation (40) becomes:(43)$$\begin{array}{c} 3D)\frac{\Delta V_{out}}{V_{in}\cdot \sigma_{v}}=8.44\times 10^{-2}\left[\frac{mV}{V\cdot MPa}\right]\\ PL)\frac{\Delta V_{out}}{V_{in}\cdot \sigma_{v}}=\;-9.67\times 10^{-3}\left[\frac{mV}{V\cdot MPa}\right] \end{array}$$

### 4.5. Increased Lateral Expansion $\upsilon_{EQ}=0.30$

In this case, the lateral expansion of the sensor will be greater than the free one, and a compressive axial deformation will be imposed; Equation (40) becomes:(44)$$\begin{array}{c} 3D)\frac{\Delta V_{out}}{V_{in}\cdot \sigma_{v}}=7.83\times 10^{-2}\left[\frac{mV}{V\cdot MPa}\right]\\ PL)\frac{\Delta V_{out}}{V_{in}\cdot \sigma_{v}}=\;-1.54\times 10^{-2}\left[\frac{mV}{V\cdot MPa}\right] \end{array}$$

### 4.6. Lateral Expansion to Match Young Concrete One $\upsilon_{EQ}=0.56$

The results of the finite element analyses shown in paragraph 5 will be compared to this case.

If a concrete with a modulus of elasticity of 25 GPa and a Poisson coefficient of 0.2 is considered, the lateral expansion it will undergo when loaded is ten times the one of the sensor as the Young modulus of ceramic is ten times bigger than concrete, and Poisson coefficients are almost equal.

This lateral expansion of concrete cannot be reached close to the sensor because of friction forces exchanged between the sensor and surrounding concrete. In a pure elastic simulation (full bond between sensor and concrete), equilibrium is reached when lateral (radial) deformation of concrete and sensor is about 2.8 times the free expansion of ceramic corresponding to a $\upsilon_{EQ}=0.56$.

In this case, the lateral expansion of the sensor will be ten times the free one; Equation (40) becomes:(45)$$\begin{array}{c} 3D)\frac{\Delta V_{out}}{V_{in}\cdot \sigma_{v}}=5.84\times 10^{-2}\left[\frac{mV}{V\cdot MPa}\right]\\ PL)\frac{\Delta V_{out}}{V_{in}\cdot \sigma_{v}}=-3.39\times 10^{-2}\left[\frac{mV}{V\cdot MPa}\right] \end{array}$$

## 5. Effect of the Presence of Soft Zones in the Sensor

One possible version of the sensor is a disk with a diameter of about 30 mm. Its gross area is, therefore:(46)$$A_{gross}=\pi 15^{2}=707\;mm^{2}$$

The areas occupied by contacts, calibration resistances, and voids in glass frit (see Figure 2d) layers are much less stiff than the solid body of the sensor.

The areas where *R*2, *R*4, *R*2*A* and *R*4*A* are placed measure:(47)$$A_{R2,R4}=4.1\times 3.6-0.5^{2}=14.1\;mm^{2}$$

The areas where *R*1*A* and *R*3*A* are placed measure:(48)$$A_{R1A,R3A}=\pi 1.7^{2}=9.1\;mm^{2}$$

The areas where contacts and calibration resistances are placed are measured:(49)$$A_{conctact}=2.14\times 24+0.86\times 12≅62\;mm^{2}$$

The net area of the sensor, that is, the area where the stress $\sigma_{v}$ passes, is:(50)$$A_{net}=A_{gross}-A_{R1A,R3A}-2\left(A_{R2,R4}+A_{conctact}\right)=707-9.1-2\left(14.1+62\right)=546\;mm^{2}$$

That means the net area is about 0.77, the gross one.

The difference in stiffness between the net area, which is very rigid, and the soft area, which is less rigid, generates on the sensor surfaces a non-uniform distribution of the stress $\sigma_{v}$, which has been considered uniformly distributed in Section 3.

## 6. Comparison with Numerical Simulations

A 3*D* finite element model of the sensor embedded inside a concrete cylinder has been done using the commercial FEA software “DIANA FEA” version 9.6 [27,28]. A brief description of the model will be presented here. The concrete cylinder has a diameter of 14 cm, and it is 8 cm tall. One eight of the specimens has been modeled in function of the symmetry conditions on two vertical planes and a horizontal one, as shown in Figure 8.

Referring to the geometry of the sensor shown in Figure 2, the finite element model has been modeled only:Bottom ceramic layer is called layer D.Bottom glass frit layer½ of the intermediate ceramic layer, called layer B

Three different degrees of refinement of the mesh have been used, as shown in Figure 9.

The dimensions and the mechanical parameters used in the finite element analysis are resumed in Table 1.

An axial uniform pressure of −10 MPa is applied to the round surfaces of the cylinder and is kept constant in time. Concrete creep (viscosity) is taken into account with a maximum creep deformation equal to two times the elastic one. Therefore, the concrete specimen undergoes increasing shortening in time under constant applied load.

The vertical stresses $\sigma_{v}$ predicted by the numerical simulations inside the glass-frit layer are shown in Figure 10. The presence of the soft areas described in paragraph 5 generates stress gradients only in small areas around the periphery of the discontinuity regions, whereas the stress is almost constant within the biggest part of the sensor body.

The zones where the 3*D* bridge is placed are subjected to a stress that is close to $\sigma_{v}$ and almost constant in time, as shown in Figure 11. A correction factor of 5 ÷ 10% should be applied to scale the stress seen by the 3*D* bridge to $\sigma_{v}$. The strains measured in correspondence with the resistors of the Wheatstone bridges for an external pressure of −10 MPa applied to the concrete specimen are presented in Table 2.

Equation (22) can then be directly applied, and therefore, the output of each bridge can be calculated using Equation (6).
(51)$$\frac{\Delta V_{out}}{V_{in}}=\frac{\Delta R}{2R+2\Delta R}≅\frac{\Delta R}{2R}$$

Obtaining the results shown in Table 3 and the following conclusions:(a)The three meshes give almost the same result, so the problem is well described, and little mesh sensitivity is found.(b)f.e.m. simulations are done using young and deformable concrete (or a mortar) to enhance the differences between concrete and ceramic and test the sensor response in difficult working conditions.(c)When load is applied, the sensitivities of the 3*D* and *PL* bridges are respectively 5.75 ÷ 6.00 × 10^−2^ and 3.75 ÷ 3.78 × 10^−2^ mV/(V MPa).(d)When viscosity is developed, the sensitivities of the 3*D* and *PL* bridges are respectively 8.08 ÷ 8.46 × 10^−2^ and 1.64 ÷ 1.68 × 10^−2^ mV/(V MPa).(e)When the viscosity is developed, the results given in point 3 are in very good agreement with the ones found by hand calculation for free natural expansion of the sensor.(f)The results shown in points b and c show that just after load application, concrete surrounding the sensor stretches it because of its lower Young modulus and higher deformability compared to ceramic. Viscosity dampens this coaction, and in the end, the sensor tends to its free natural expansion.(g)A response constant in time is found by combining the two bridges with the equation 3*D* − 1.14 *PL*.(h)The response of the sensor is related to virtual stress, which is 5 ÷ 10% bigger than the one applied to the concrete specimen, as seen in Figure 10. A reduction factor should then be applied to the output, obtaining a combination of the two bridges of 0.95 × (3*D* − 1.10 *PL*).

## 7. Results Discussion

The results of the closed-form calculations are summarized for the five load cases in Table 4 and compared with finite element simulation results. The same data is plotted in Figure 12.

A constant response of the sensor despite the variation of lateral confinement $\upsilon_{EQ}$ is found by combining the closed-form calculations of the two bridges with the equation 3*D* − 1.07 *PL* or the f.e.m calculations with the equation 0.95 × (3*D* − 1.10 *PL*).

The output of the planar and 3*D* bridges are also plotted in Figure 12 (blue and orange curves) per 1 MPa (compression is negative). The linear combinations of the two bridges, 3*D* − 1.07 *PL* and 0.95 × (3*D* − 1.10 *PL*), plotted in greed and turn out to be perfectly constant, demonstrating the complete indifference of the sensor to confinement or to viscosity and so demonstrating it can be used to measure pressures inside viscoelastic materials.

The sensitivity of the bridges is slightly different from the ones measured during laboratory tests done in concrete. A mean sensitivity for the *PL* and 3*D* bridges, respectively, of −1.56 × 10^−2^ and 7.11 × 10^−2^ was measured in a laboratory compression test. This result can be compared with the closed-form solution obtained with $\upsilon_{EQ}=0.30$. The comparison makes sense, considering that common concrete tested in the laboratory is stiffer than the one used in the f.e.m. model presented in this paper. The sensitivity of the sensor tested in the laboratory may also be slightly lower than the theoretical one because sensors were not perfectly aligned inside laboratory specimens because of concrete casting and vibrations.

## 8. Conclusions

This study presents a new sensor that can be embedded in concrete or masonry structures during construction to measure the stress orthogonal to its faces. 

The measurement of stress within a solid body is usually obtained indirectly by measuring strains and then converting them into stresses by knowing in advance the constitutive equation of the material to be measured. This conversion is simple and reliable for linear elastic materials. Major difficulties occur trying to measure stress within a structure in which the material mechanical characteristics are neither uniform in space nor constant over time and are generally not precisely known in advance, as in the case of all cementitious materials like concrete, mortars, and masonry. In addition, concrete and mortar are viscous materials that undergo big strain variations (sometimes bigger than 200%) under constant load because of creep. Therefore, no direct relation can be drawn between the measure of the strain and the level of stress.

For these reasons, the sensor presented in this study is developed to measure the internal stress of the structure directly. It exploits the piezo resistivity of thick ink film based on ruthenium oxide, and it is insensitive with respect to the stiffness of the embedding material and the variation of the surrounding material properties (like hardening, shrinkage, or creep).

Different boundary conditions of the sensor are analytically investigated in closed form, simulating its working conditions outside and inside a concrete matrix. One analytically simulated condition is also investigated with the support of a non-linear f.e.m. model where the sensor and the concrete matrix surrounding it are modeled. An excellent correspondence between analytical results calculated in closed form and numerical simulation is found. 

A thorough experimental campaign has been accomplished by the authors testing tens of sensors in different operating conditions, confirming the theoretical results presented in this paper.

## 9. Patents

Bertagnoli G. (Inventor); Safecertifiedstructure Tecnologia s.r.l. (Applicant). Patent: Method and investigation device for measuring stresses in an agglomerate structure, International Publication Number: WO 2017/178985 Al, Published: 19 October 2017.Guidetti, E., Abassi Gavarti, M., Caltabiano, D. and Bertagnoli, G. (inventors); STMicroelectronics S.r.l. (Applicant). Patent: Stress Sensor for Monitoring the Health state of fabricated Structures, such as constructions, Buildings, Infrastructures, and the like. European Patent Specification EP 3 392 637 B1. Published 27 November 2019.

## Figures and Tables

**Figure 1 sensors-24-00599-f001:**
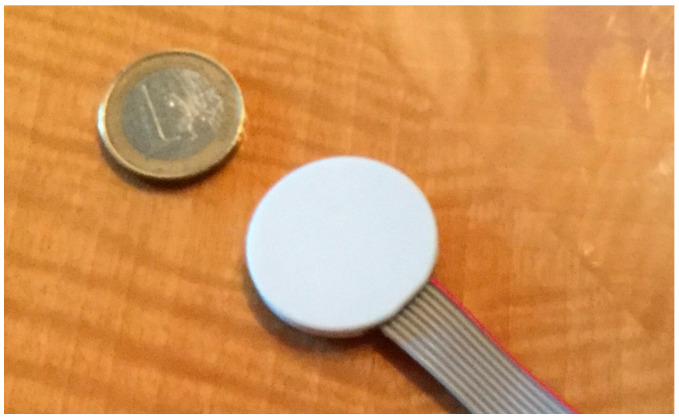
Stress sensor [21].

**Figure 2 sensors-24-00599-f002:**
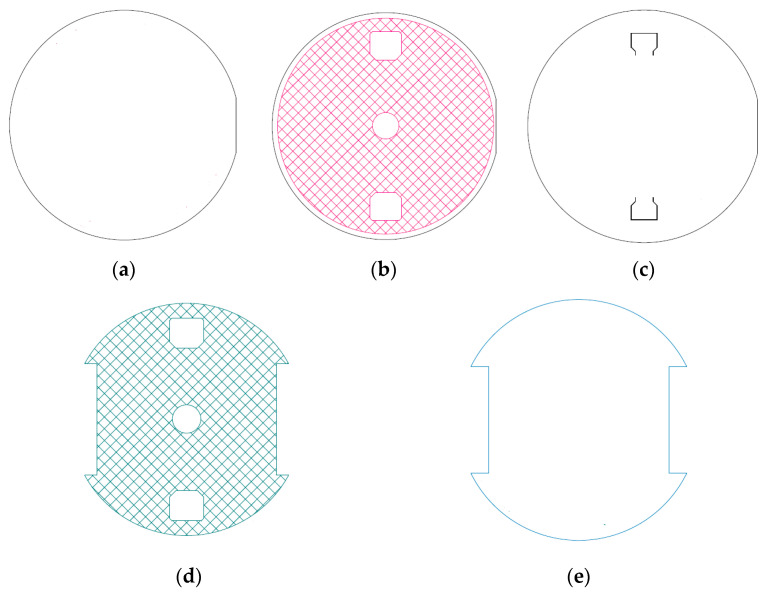
Layers inside sensor: (**a**) Bottom ceramic layer (thickness 1.5 ÷ 2.0 mm); (**b**) Bottom glass frit—between bottom and intermediate ceramic layer (thickness 0.010 ÷ 0.020 mm); (**c**) Intermediate ceramic layer (thickness 0.3 ÷ 1.0 mm); (**d**) Top glass frit between intermediate and top ceramic layer (thickness 0.04 ÷ 0.06 mm); (**e**) Top ceramic layer (thickness 1.5 ÷ 2.0 mm).

**Figure 3 sensors-24-00599-f003:**
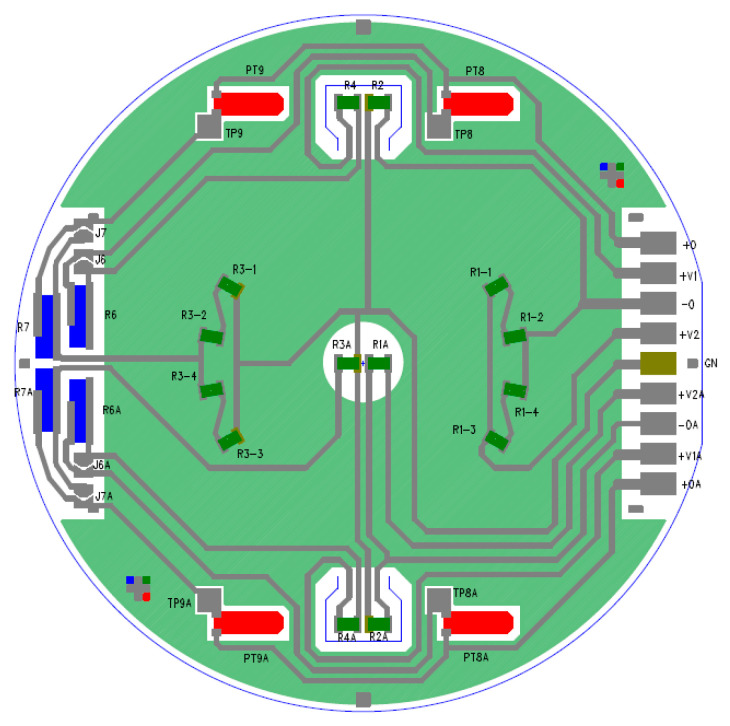
Electrical scheme of Wheatstone bridges.

**Figure 4 sensors-24-00599-f004:**
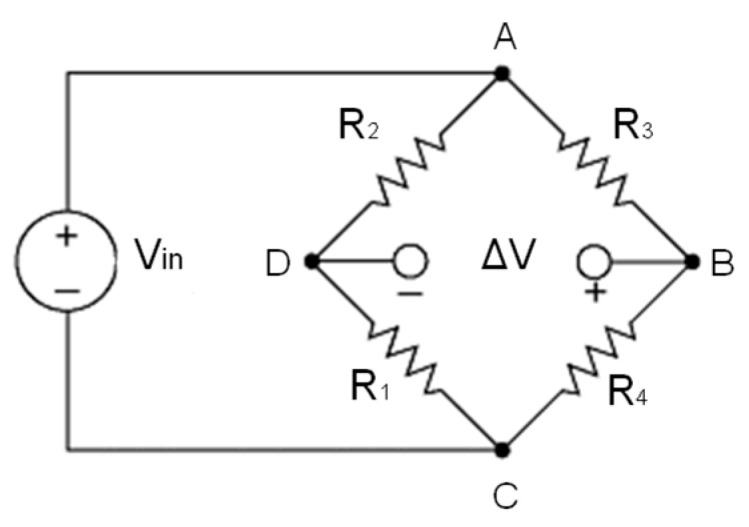
Wheatstone bridge.

**Figure 5 sensors-24-00599-f005:**
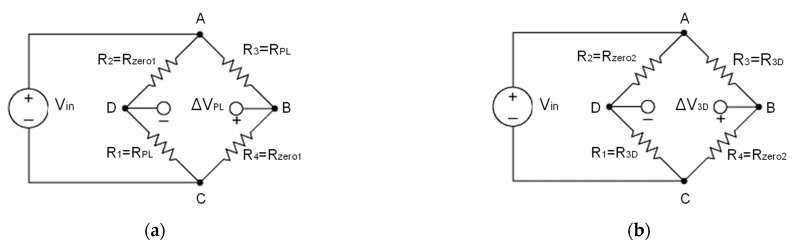
Wheatstone bridges: (**a**) *PL* bridge; (**b**) 3*D* bridge.

**Figure 6 sensors-24-00599-f006:**
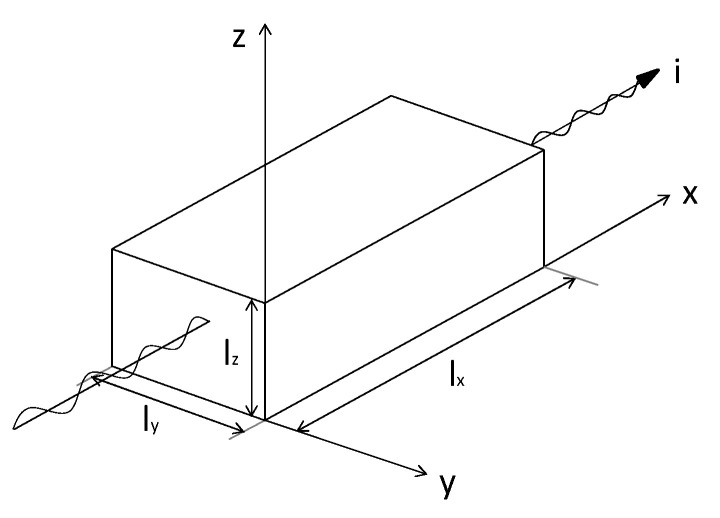
Local reference system for resistors.

**Figure 7 sensors-24-00599-f007:**
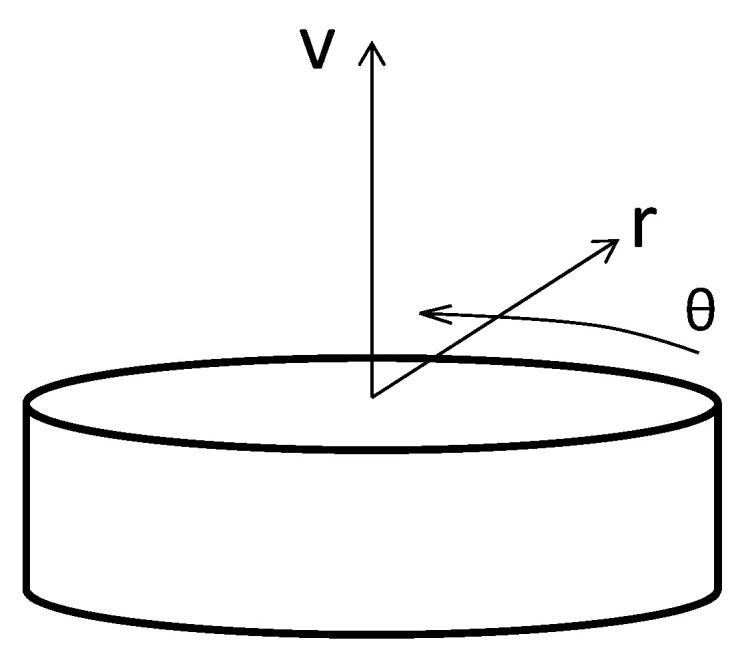
Global reference system for the sensor.

**Figure 8 sensors-24-00599-f008:**
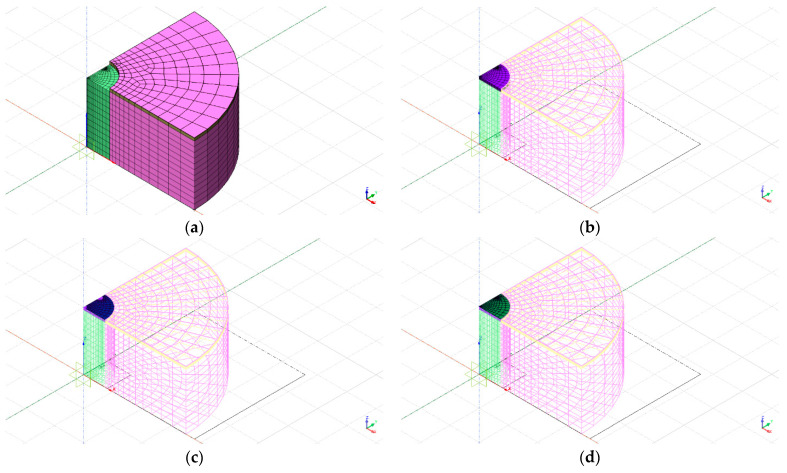
f.e.m. of the sensor embedded in concrete: (**a**) Concrete mesh; (**b**) Ceramic layer D mesh; (**c**) Glass-frit mesh; (**d**) Ceramic layer B mesh.

**Figure 9 sensors-24-00599-f009:**
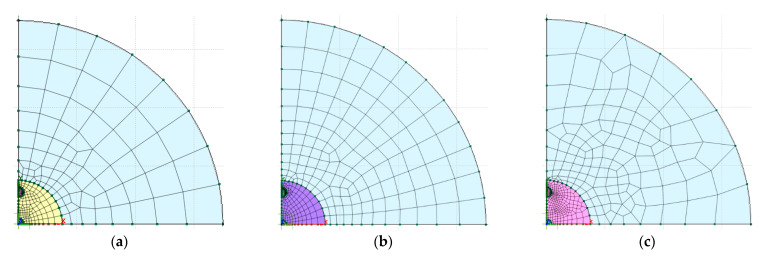
Mesh refinement: (**a**) 5352 nodes; (**b**) 7039 nodes; (**c**) 10,492 nodes.

**Figure 10 sensors-24-00599-f010:**
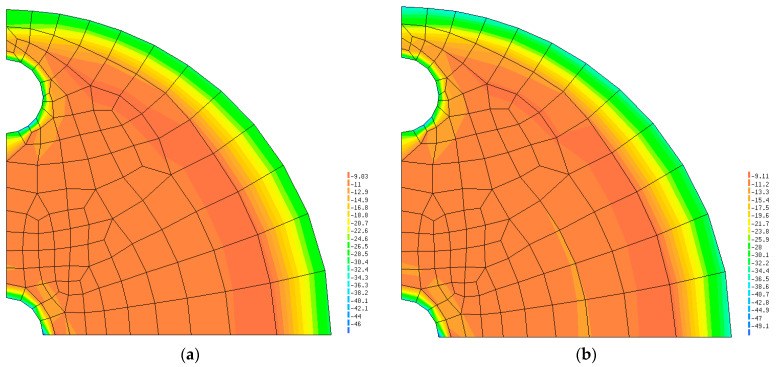
Vertical stresses in the glass-frit layer: (**a**) Just after load; (**b**) Full viscosity developed.

**Figure 11 sensors-24-00599-f011:**
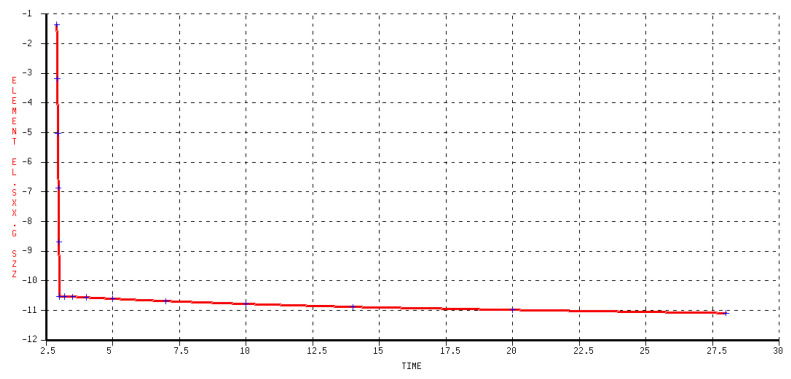
Vertical stress [MPa] in 3*D* bridge zone in function of time (creep effect).

**Figure 12 sensors-24-00599-f012:**
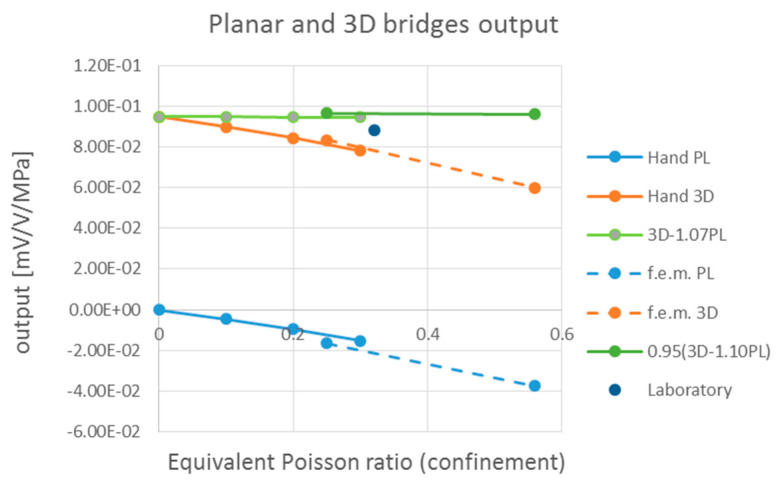
Planar and 3*D* bridge output is in the function of confinement.

**Table 1 sensors-24-00599-t001:** Input data of f.e.m. analysis.

Element	Material	Radius [mm]	Thickness [mm]
Concrete (mortar) cylinder	E variable in time E (28 days) = 23.5 GPa $\nu_{con}=0.2$	70	40
Ceramic layer D	E = 250 GPa $\nu_{cer}=0.22$	15	1.7
Glass-frit	E = 78 GPa $\nu_{cer}=0.22$	15	0.045
Ceramic layer B	E = 250 GPa $\nu_{cer}=0.22$	15	0.1

**Table 2 sensors-24-00599-t002:** Strains in resistors.

		Just after Load			Full Concrete Viscosity Developed		
		$\varepsilon_{x}$	$\varepsilon_{y}$	$\varepsilon_{z}$	$\varepsilon_{x}$	$\varepsilon_{y}$	$\varepsilon_{z}$
Mesh 5352	*R*3*D*	2.60 × 10^−5^	2.60 × 10^−5^	−1.30 × 10^−4^	1.10 × 10^−5^	8.50 × 10^−6^	−1.25 × 10^−4^
	*RPL*	3.15 × 10^−5^	3.10 × 10^−5^	−1.75 × 10^−5^	1.40 × 10^−5^	1.30 × 10^−5^	−7.56 × 10^−6^
Mesh 7039	*R*3*D*	2.50 × 10^−5^	2.60 × 10^−5^	−1.32 × 10^−4^	9.00 × 10^−6^	8.50 × 10^−6^	−1.27 × 10^−4^
	*RPL*	3.14 × 10^−5^	3.06 × 10^−5^	−1.74 × 10^−5^	1.44 × 10^−5^	1.30 × 10^−5^	−7.67 × 10^−6^
Mesh 10,492	*R*3*D*	2.55 × 10^−5^	2.60 × 10^−5^	−1.31 × 10^−4^	9.00 × 10^−6^	8.00 × 10^−6^	−1.28 × 10^−4^
	*RPL*	3.15 × 10^−5^	3.07 × 10^−5^	−1.74 × 10^−5^	1.43 × 10^−5^	1.35 × 10^−5^	−7.78 × 10^−6^

**Table 3 sensors-24-00599-t003:** Bridges output from finite element simulation.

	Just after Load			Full Viscosity Developed		
	DR/R	*V_out_/V_in_*	3*D*-aPL	DR/R	*V_out_/V_in_*	3*D*-aPL
	[-]	[mV/(V MPa)]	[mV/(V MPa)]	[-]	[mV/(V MPa)]	[mV/(V MPa)]
Mesh 5352	−1.15 × 10^−3^	5.75 × 10^−2^	1.01 × 10^−1^	−1.62 × 10^−3^	8.08 × 10^−2^	9.95 × 10^−2^
	7.56 × 10^−4^	−3.78 × 10^−2^		3.27 × 10^−4^	−1.64 × 10^−2^	
Mesh 7039	−1.20 × 10^−3^	5.99 × 10^−2^	1.03 × 10^−1^	−1.67 × 10^−3^	8.34 × 10^−2^	1.02 × 10^−1^
	7.50 × 10^−4^	−3.75 × 10^−2^		3.33 × 10^−4^	−1.66 × 10^−2^	
Mesh 10,492	−1.17 × 10^−3^	5.87 × 10^−2^	1.02 × 10^−1^	−1.69 × 10^−3^	8.46 × 10^−2^	1.04 × 10^−1^
	7.53 × 10^−4^	−3.76 × 10^−2^		3.37 × 10^−4^	−1.68 × 10^−2^	

**Table 4 sensors-24-00599-t004:** Summary of theoretical results.

Closed Form Calculation				f.e.m. Calculation		
Case	Equivalent Confinement Poisson Ratio	*PL* Bridge Output $\frac{mV}{V\cdot MPa}$	3*D* Bridge Output $\frac{mV}{V\cdot MPa}$	*PL* Bridge Output $\frac{mV}{V\cdot MPa}$	3*D* Bridge Output $\frac{mV}{V\cdot MPa}$	Case
Nil lateral expansion	$\upsilon_{EQ}=0.00$	−0.00 × 10^0^	9.48 × 10^−2^			
Confined lateral expansion	$\upsilon_{EQ}=0.10$	−4.58 × 10^−3^	8.99 × 10^−2^			
Free lateral expansion	$\upsilon_{EQ}=0.20$	−9.67 × 10^−3^	8.44 × 10^−2^	−1.64 × 10^−2^	8.30 × 10^−2^	Full viscosity developed
Increased lateral expansion	$\upsilon_{EQ}=0.30$	−1.54 × 10^−2^	7.83 × 10^−2^			
Increased lateral expansion to match fem simulations	$\upsilon_{EQ}=0.56$	−3.39 × 10^−2^	5.84 × 10^−2^	−3.76 × 10^−2^	5.87 × 10^−2^	Young concrete just after load application

## Data Availability

Data available on request due to restrictions (e.g., privacy, legal or ethical reasons).

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
