# Peer review of "Ceramic Stress Sensor Based on Thick Film Piezo-Resistive Ink for Structural Applications"

_sensors, 2024, doi:10.3390/s24020599_

Round 1

Reviewer 1 Report

Comments and Suggestions for Authors

The authors need to improve their presentation and make their statement more concise. The following is some examples on the specific problems of their presentation.

1. The authors made the error at the beginning of the manuscript, which is the title. The title is:Ceramic Stress Sensor Based on Tick film Piezo-Resistive Ink for Structural Applications. While, there is no such thing called tick film. And the words of tick film also appear in reference 13. I actually looked up this reference and found that the correct spelling is thick film. Furthermore, the journal name of reference 13 is also wrong, which should be Sensors and Actuators A: Physical 

2. The illustration of Figure 7: Global referring system for the sensor.  The word referring should be reference. Furthermore, I do not understand the word global. Should the global reference system here just mean the inertial reference system?

3. The authors presented a lot of formulae. However, quite some of them are just the basics of elasticity, electricity and strain gauge principle. For a journal paper, there is no need to give so many basis formulae.   

4. There are ten paragraphs in Conclusion. Your main contribution is to use some fundamental knowledges to design/compute a

Comments on the Quality of English Language

It is not good and they even mispelled the word thick as tick in their manusrcipt title.

Author Response

Reviewer 1

Dear Reviewer 1, thank you for your time and valuable suggestions, we will answer to your comments in the following.

The authors need to improve their presentation and make their statement more concise. The following is some examples on the specific problems of their presentation.
  1. The authors made the error at the beginning of the manuscript, which is the title. The title is: Ceramic Stress Sensor Based on Tick film Piezo-Resistive Ink for Structural Applications. While, there is no such thing called tick film. And the words of tick film also appear in reference 13. I actually looked up this reference and found that the correct spelling is thick film. Furthermore, the journal name of reference 13 is also wrong, which should be Sensors and Actuators A: Physical
Answer to point 1. it is a typo; we apologise for this error. both the title and reference 13 have been updated.
  1. The illustration of Figure 7: Global referring system for the sensor. The word referring should be reference. Furthermore, I do not understand the word global. Should the global reference system here just mean the inertial reference system?
Answer to point 2. Right there is a typo, the correct word is reference, and it has been changed. Yes it refers to the inertial reference system, but in polar coordinates.
  1. The authors presented a lot of formulae. However, quite some of them are just the basics of elasticity, electricity and strain gauge principle. For a journal paper, there is no need to give so many basis formulae.
Answer to point 3.   It is true that there are probably too many formulae, but the authors believe that the formulae given are essential in order to follow the procedure without losing parts.
  1. There are ten paragraphs in Conclusion. Your main contribution is to use some fundamental knowledges to design/compute.
Answer to point 4. Conclusions were revised and reorganised.

Reviewer 2 Report

Comments and Suggestions for Authors

This manuscript proposes a design of a ceramic stress sensor that can be embedded in concrete or masonry structures. The manuscript is recommended for publication after major revision. The problems should be addressed shown as follows:

Comment 1.Insufficient description of recently published papers in ceramic stress sensors.

This work presents the design of a ceramic stress sensor. The author analyzes the basic background information of solutions frequently used for investigating concrete or masonry structures. However, it is important to refer to recent studies on ceramic stress sensors.

Comment 2. It is a bit unclear how the authors determine the temperature compensation in the Abstract. However, the initial value of the sensor in the temperature test was quite different from that in experiments, and the parameters of the temperature test environment should be provided and analyzed.

There are several spelling/grammatical errors and missing words in sentences. The authors need to review the final manuscript carefully.

Comment 3a. Correct the spelling error in the title, changing 'Tick film' to 'Thick film'.

Comment 3b. Ensure uniform formatting of subscripts in formula symbols.

Comment 3c. Pay attention to the correctness of units in equations, such as the length unit in Eq. (10).

Comments on the Quality of English Language

Moderate editing of English language required

Author Response

Reviewer 2

Dear Reviewer 2, thank you for your time and valuable suggestions, we will answer to your comments in the following.

This manuscript proposes a design of a ceramic stress sensor that can be embedded in concrete or masonry structures. The manuscript is recommended for publication after major revision. The problems should be addressed shown as follows:

Comment 1. Insufficient description of recently published papers in ceramic stress sensors.

This work presents the design of a ceramic stress sensor. The author analyzes the basic background information of solutions frequently used for investigating concrete or masonry structures. However, it is important to refer to recent studies on ceramic stress sensors.

Answer to point 1. In section 2, the authors tried to improve the references on recent studies on ceramic stress sensors.

Comment 2. It is a bit unclear how the authors determine the temperature compensation in the Abstract. However, the initial value of the sensor in the temperature test was quite different from that in experiments, and the parameters of the temperature test environment should be provided and analyzed.

There are several spelling/grammatical errors and missing words in sentences. The authors need to review the final manuscript carefully.

Answer to point 2. In the introduction, the authors now present some information about the thermal compensation process, nevertheless it is not the main topic of the paper, therefore it has been removed from the abstract.

Comment 3a. Correct the spelling error in the title, changing 'Tick film' to 'Thick film'.

Answer to point 3a. Checked, it was a typo.

Comment 3b. Ensure uniform formatting of subscripts in formula symbols.

Answer to point 3b. Checked.

Comment 3c. Pay attention to the correctness of units in equations, such as the length unit in Eq. (10).

Answer to point 3c. Checked, it was a typo, the lengths were given in micrometres and not nanometres.

Reviewer 3 Report

Comments and Suggestions for Authors

The overall research idea is clear and the research content is well verified by analogue simulation, which is suitable for revision and publication. The introduction section should be a description of the progress and challenges of Ceramic Stress Sensor research and proposed design solutions, however the current form is disorganised. Further adjustments are recommended. Secondly, the conclusion of a research work is too lengthy, and it is recommended to condense and summarise the key information so that readers can grasp the important progress made in this work.

Comments on the Quality of English Language

Moderate adjustment

Author Response

Reviewer 3

Dear Reviewer 3, thank you for your time and valuable suggestions, we will answer to your comments in the following.

The overall research idea is clear and the research content is well verified by analogue simulation, which is suitable for revision and publication. The introduction section should be a description of the progress and challenges of Ceramic Stress Sensor research and proposed design solutions, however the current form is disorganised. Further adjustments are recommended. Secondly, the conclusion of a research work is too lengthy, and it is recommended to condense and summarise the key information so that readers can grasp the important progress made in this work.

The authors have reorganised and improved both the introduction and conclusion of the article.

Reviewer 4 Report

Comments and Suggestions for Authors

The work presented here concerns the description of the electromechanical characteristics of a membrane transducer dedicated to compressive force measurement. The transducer uses elements that exhibit a piezoresistive effect. On the basis of the constitutive equations, the Authors provided close solutions, allowing the determination of normal stresses in the transducer, formed under the influence of an external load. The results obtained from the analytical solutions were compared with those obtained by numerical testing (FEM). Satisfactory agreement of the results was obtained.

In my opinion, the work is interesting, and the proposed solution can find practical application. I did not find any significant factual errors in this work. However, before publishing the manuscript, the authors should provide information about:

-whether the sensor can be used to test a structure loaded both statically and dynamically/cyclically;

-whether the analytical solution accounts for possible changes in resistance as a fixed effect after deformation;

-the measurement range and accuracy;

In addition, the manuscript includes drawings/photographs published in other works of the authors, so it would have been appropriate to indicate their source.

Round 2

Reviewer 1 Report

Comments and Suggestions for Authors

The authors actually refused to revise according to my suggestion of No. 3 point, which is a major revision and not necessary as far as I can see. And I still insist on the No. 3 point. 

Comments on the Quality of English Language

The English presentation can still be improved.

Reviewer 2 Report

Comments and Suggestions for Authors

There are still some sentences and text descriptions that lack formality. For instance, the use of "the authors" as the subject in lines 50 and 55 is not recommended.

Comments on the Quality of English Language

Minor editing of English language required

Author Response

We tried to improve English throughout the paper.

Reviewer 3 Report

Comments and Suggestions for Authors

The authors have made the required revisions, and I recommend that it be accepted in its current form for publication.

Author Response

No revision was asked by Reviewer 3.

We tried to improve English throughout the paper and followed the requests of reviewer 1.